# Non-Marked Hypoechogenic Nodules: Multicenter Study on the Thyroid Malignancy Risk Stratification and Accuracy Based on TIRADS Systems Comparison

**DOI:** 10.3390/medicina58020257

**Published:** 2022-02-09

**Authors:** Peteris Prieditis, Maija Radzina, Madara Mikijanska, Mara Liepa, Kaspars Stepanovs, Giorgio Grani, Cosimo Durante, Livia Lamartina, Pierpaolo Trimboli, Vito Cantisani

**Affiliations:** 1Radiology Research Laboratory, Riga Stradins University, LV-1007 Riga, Latvia; peteris.prieditis@rsu.lv (P.P.); mara.liepa@rsu.lv (M.L.); 2Diagnostic Radiology Institute, Pauls Stradins Clinical University Hospital, LV-1002 Riga, Latvia; madara.mikijanska@stradini.lv (M.M.); kstepanovs@gmail.com (K.S.); 3Department of Radiological, Anatomopathological and Oncological Sciences, Sapienza University of Rome, 00100 Rome, Italy; info@giorgiograni.it (G.G.); cosimo.durante@uniroma1.it (C.D.); livialamartina@gmail.com (L.L.); vito.cantisani@uniroma1.it (V.C.); 4Clinic for Endocrinology and Diabetology, Lugano Regional Hospital, Ente Ospedaliero Cantonale, 6900 Lugano, Switzerland; pierpaolo.trimboli@eoc.ch; 5Faculty of Biomedical Sciences, Università Della Svizzera Italiana, 6900 Lugano, Switzerland

**Keywords:** thyroid nodule, TIRADS, aspiration biopsy, ultrasound, thyroid cancer

## Abstract

*Background and Objectives*: The aim of the study was to evaluate the predictive value of the ultrasound criterion “non-marked hypoechogenicity” for malignancy and to determine whether classification of these nodules as TIRADS 3 could improve the overall accuracy of consequently adjusted M-TIRADS score. *Materials and Methods*: A total of 767 patients with 795 thyroid nodules were subject to ultrasonography examination and ultrasound-guided fine needle aspiration biopsy. Nodules were classified by Kwak TIRADS and modified (M-TIRADS) categories 4A, 4B, and 5 according to number of suspicious US features (marked hypoechogenicity, microlobulated or irregular margins, microcalcifications, taller-than-wide shape, metastatic lymph nodes). Non-marked hypoechoic nodules were classified as TIRADS 3. *Results*: Thyroid nodules were classified as TIRADS 2, 3, 4A, 4B, and 5 in 14.5, 57.5, 14.2, 8.1, and 5.7%, respectively. Only histopathologic results (125 nodules underwent surgery) and highly specific cytology results (Bethesda II, VI) were accepted as a standard of reference, forming a sub-cohort of 562/795 nodules (70.7%). Malignancy was found in 7.7%. Overall, M-TIRADS showed sensitivity/specificity of 93.02/81.31%, and for PPV/NPV, these were 29.2/99.29%, respectively (OR—18.62). Irregular margins showed the highest sensitivity and specificity (75.68/93.74%, respectively). In TIRADS 3 category, 37.2% nodules were isoechoic, 6.6% hyperechoic, and 52.2% hypoechoic (there was no difference of malignancy risk in hypoechoic nodules between M-TIRADS and Kwak systems—0.9 vs. 0.8, respectively). Accuracy of M-TIRADS classification in this cohort was 78.26% vs. 48.11% for Kwak. *Conclusions*: The non-marked hypoechoic nodule pattern correlated with low risk of malignancy; classification of these nodules as TIRADS 3 significantly improved the predictive value and overall accuracy of the proposed M-TIRADS scoring with malignancy risk increase in TIRADS 4 categories by 20%; and no significant alteration of malignancy risk in TIRADS 3 could contribute to reducing overdiagnosis, obviating the need for FNA.

## 1. Introduction

With a large regional variation, 7–15% of all thyroid nodules are malignant, depending on age, sex, radiation exposure history, family history, and other factors [1]. The most common entity is differentiated thyroid cancer—papillary ad follicular thyroid carcinoma. Thyroid cancer incidence has been nearly tripled since last century, while mortality remained stable [2,3,4]. An increase of incidence may be partially explained by over-diagnosis due to the widespread use of ultrasound (US) examinations and partially by the increased number of biopsies. Moreover, more thyroid nodules are accidentally detected due to the increase of imaging techniques [5]. Currently the method of choice for treatment is surgery, but alternatively, active surveillance may be justified in small papillary thyroid carcinomas [6,7,8]. It is a challenge to distinguish clinically significant malignant thyroid nodules requiring surgery from benign thyroid nodules that require long-term observation [9].

Overdiagnosis and overtreatment of thyroid nodules have even been called “epidemic” by some authors, regarding consequent increasing financial costs, surgical complications, and need for lifetime thyroid replacement therapy. The Thyroid Imaging Reporting and Data System (TIRADS) was introduced in 2009 in order to achieve a systematic evaluation of malignancy risk and to reduce the number of biopsies [10]. It has been modified, improved, and supplemented to enhance the accuracy of the scoring. The main limitations are the relative low specificity of different ultrasonographic features with limited positive and negative predictive values for malignancy.

Hypoechogenicity is one of the main but unspecific features of thyroid nodules on US [11]. The positive predictive value of this feature is weak (PPV and OR ranging from 3.57 to 6.63) [12,13]. In favor of making this criterion more selective, a lower echogenicity compared to muscle tissue termed “marked hypoechogenicity” has been introduced as a marker of increased risk of malignancy in solid nodules. It is reported that up to 55% of benign nodules appear hypoechoic compared to thyroid parenchyma, making nodule non-marked hypoechogenicity less specific, especially for sub-centimeter size [1]. However, several studies report that the majority of thyroid nodules are hypoechogenic, both benign and malignant [14,15,16].

The goal of the present study was to evaluate the diagnostic accuracy of the current TIRADS system. It was hypothesized that in consistency with the assumption of a higher malignancy in markedly hypoechogenic nodules, all remaining hypoechogenic lesions with equal or higher echogenicity compared to muscle tissue (“non-marked hypoechogenic nodules”) would not be associated with an increased risk of malignancy and their classification as grade 3 would significantly improve the predictive value of the TIRADS classification.

## 2. Materials and Methods

### 2.1. Study Design

This diagnostic multicenter study with pro- and retrospective design was carried out at three medical institutions in Latvia. Within the study period from 2015 to 2017, all consecutive patients with suspicion of thyroid nodules were subject to a standardized ultrasound examination following the standard clinical pathways. All patients with an appropriate clinical indication for fine needle aspiration biopsy (FNA) as indicated following the institutional standards were included in the study. Only lesions with conclusive results from FNA were further analyzed. All participants were treated according to the institutional standards, and their management was not influenced by the study protocol, except for the standardized documentation of the ultrasound examination. Any further investigation was based on the retrospective analysis of the clinical reports. The study was approved by the institutional review boards and the responsible ethics committee; informed written consent for the documentation of the ultrasound examination and the retrospective, anonymized analysis of the clinical reports was waived according to the local regulations.

Prior to the study, a consensus meeting was held to establish uniform methodology and to reach interpretation agreement among 3 institutions by use of series of the cases. Consensus meeting was multidisciplinary—endocrinology, radiology, cytopathology, and surgery specialists were involved. No US system performance nor interobserver comparison was performed; therefore, these were not included in the statistical analysis.

### 2.2. Acquisition of Data

In addition to the regular clinical report, the results of the ultrasound examination were documented with a standard form. The standard form included patient data, suspicious US features, and all necessary components for M-TIRADS classification. In case of a multinodular goiter, the radiologist chose the most suspicious nodule or the largest one of any equally suspicious nodules. Upon retrospective review of all patient ultrasound examinations at the end of the study, firstly, we used our system M-TIRADS to characterize thyroid nodules with subsequent re-classification of the same nodule parameters into the Kwak TIRADS system [13]. It was performed by the same person who performed the initial classification.

### 2.3. US Examination

US scanning was performed with three different US systems—one in every institution (Epiq 5 (Philips, Amsterdam, Netherlands) and Aplio 800 system (Canon, Tokyo, Japan)) using a linear array transducer at 5–18 MHz. The examinations were performed by one of three board-certified radiologists with 5 to 28 years of experience in thyroid imaging and biopsy technique. All US guided FNAs were performed and documented in the same session by the same radiologist. Nodule size, position, internal component, echogenicity, margins, calcification, and shape were noted. Thyroid parenchyma and neck lymph nodes were also explored.

### 2.4. Modified TIRADS Classification

Each nodule was classified by two systems: Kwak and new modified Kwak TIRADS (M-TIRADS) score according to the number of suspicious US features by the radiologist performing the FNA. Before study initiation, the consensus meeting agreement was drawn regarding evaluation of criteria to unify interpretation principles for the classification. These suspicious US features were marked hypoechogenicity, microlobulated or irregular margins, microcalcifications, taller-than-wide shape, and metastatic lymph nodes. The TIRADS 2 group corresponded to benign nodule: simple cyst, spongiform nodule, homogeneous hyperechoic nodules, chronic thyroiditis, and solitary macrocalcification. The TIRADS 3 group likely corresponded to benign nodules without any suspicious US features; however, as it was the only modification with respect to Kwak TIRADS, in this group, we also included mildly hypoechoic nodules (Figure 1, Figure 2 and Figure 3). The TIRADS 4A nodules had one US feature suspicious for malignancy; TIRADS 4B had two US features, and TIRADS 5 had three or more US features such as marked hypoechogenicity (Figure 4). If metastatic lymph nodes were found, the nodule was classified as TIRADS 5, disregarding any other suspicious US features.

### 2.5. Fine Needle Aspiration (FNA)

US-guided FNA biopsy was performed with a 23-gauge needle. Materials were expelled onto glass slides, smeared, and submitted for cytology [17]. All samples were analyzed by a specialized cytopathologist with at least 5 years of experience in thyroid pathology and classified into Bethesda classification: non-diagnostic (category I), benign (category II), atypia of undetermined significance or follicular lesion of undetermined significance (category III), follicular neoplasm or suspicious for a follicular neoplasm (category IV), suspicious for malignancy (category V), or malignant (category VI) [1,18,19]. We analyzed cytopathology as routine cases, and no special selection was performed. This study did not include the comparison of interobserver accuracy.

### 2.6. Standard of Reference

As far as is available, histological diagnosis has been the gold standard for patients submitted to thyroid removal. In patients with no indication for surgery, only nodules with Bethesda II or Bethesda VI cytological report were included, since the probability of false negatives or positives of these categories is expected at less than 3% [2].

### 2.7. Statistical Analysis

Statistical analysis was performed using the software IBM SPSS 13.0 (SPSS Inc., Chicago, IL, USA). M-TIRADS category 2 and 3 were estimate as “test negative”, and M-TIRADS 4 and 5 as “test positive”. Sensitivity, specificity, positive predictive value (PPV), negative predictive value (NPV), and odds ratio (OR) were calculated for each suspicious US feature and M-TIRADS as diagnostic test with reference to FNA cytology. The same evaluations were performed with reference to surgical histology for patients undergoing thyroidectomy. Comparison of suspicious US features amid benign and malignant nodules were performed using the chi-squared test. The risk of malignancy of each M-TIRADS group was determined. *P*-value < 0.05 was considered indicative of a statistically significant difference.

## 3. Results

A total of 795 nodules in 767 patients were subject to FNA during the study period, including 673 females and 94 males, mean age 55.5 ± 14.1 years with a range of 8 to 87. These 795 nodules were classified as TIRADS 2, 3, 4A, 4B, and 5 in 14.5, 57.5, 14.2, 8.1, and 5.7%, respectively. FNA was successful in 562 nodules with cytological results according to Bethesda II and Bethesda VI in 519 (64.8%) and 43 (5.2%) nodules, respectively. The other nodules were excluded due to FNA report of non-diagnostic (*n* = 171, 21.5%), Bethesda III (*n* = 19, 2.4%), Bethesda IV (*n* = 10, 1.3%), or Bethesda V (*n* = 33, 4.2%). In six cases, there were only histological results; therefore, we divided these nodules as benign or malignant on the basis of the results. Among all patients, 119 underwent surgery, and histopathology was obtained from 125 nodules.

### 3.1. Performance of M-TIRADS and K-TIRADS Using FNA as Reference Standard

In the 562 nodules with benign and malignant cytology, we recorded 102 TIRADS 2 with a cancer prevalence of 0.0%, 323 TIRADS 3 with a cancer prevalence of 0.9%, 77 TIRADS 4A with a cancer prevalence of 6.5%, 32 TIRADS 4B with a cancer prevalence of 37.5%, and 28 TIRADS 5 with a cancer prevalence of 82.1% (Figure 5).

Overall, M-TIRADS showed an excellent sensitivity of 93.02% and a NPV of 99.29%, with a significant OR of 18.62. Table 1 details the results of predictive tests of M-TIRADS and performance of single suspicious US features for malignancy.

A series of 736 nodules could be re-classified according to original Kwak TIRADS, and these had a Bethesda II or Bethesda VI FNA report. Comparison of M-TIRADS (*n* = 562) and Kwak TIRADS (*n* = 526) is shown in Table 2.

Overall performance of Kwak TIRADS was as follows: sensitivity—96.97% (95% CI 84.24–99.92%); specificity—55.58% (95% CI 51.07–60.02%); PPV—12.75% (95% CI 11.52–14.09%); NPV—99.64% (95% CI 97.54–99.95%); accuracy—58.17% (CI 95% 53.83–62.43%).

Both TIRADS systems were analyzed by chi-squared test—for M-TIRADS, it was 86.25, and for Kwak TIRADS, 45.23 (*p* < 0.001), showing a statistically significant marked difference and weak correlations between cytology results for M-TIRADS (rs 0.42) and Kwak TIRADS (rs 0.22), *p* = 0.0001. Area under the curve was 0.70 (95% CI 0.634–0.776) for Kwak TIRADS and 0.86 (95% CI 0.800–0.913) for M-TIRADS in ROC curve analysis, *p* = 0.0001 (Figure 1), which indicated that only one US sign—non-marked hypoechogenicity based different grading in the TIRADS system—was able to improve accuracy (Figure 6).

### 3.2. Results of M-TIRADS in the Subgroup of Surgically Treated Patients

Among those 125 nodules with histologic diagnosis, 51 were malignant and 70 benign. In four cases, FNA biopsy was taken from a second, benign nodule aside from another malignant nodule in the thyroid gland. In all four cases, the histopathology revealed papillary carcinoma. Overall, 70 benign lesions (43 nodular goiters, 19 follicular adenomas, 4 oncocytomas, and 4 nodular thyroiditis) and 51 cancers (25 papillary carcinomas, 17 follicular variants of papillary carcinomas, 3 follicular carcinomas, 4 medullary carcinomas, 1 lymphoma, and 1 anaplastic carcinoma) were found. In the one case of lymphoma, M-TIRADS category was 2 and cytology result was benign. Table 3 shows the summary of M-TIRADS category, cytology results, and histological results in our cohort. The comparison of histology and cytology results based on M-TIRADS groups showed remarkable data—histology revealed more malignant cases than cytology. The risk of malignancy according to histology results for M-TIRADS categories 3 to 5 was 17.7, 58.3 and 76%, respectively. Only seven nodules were classified as M-TIRADS 2, with 1/7 being lymphoma. Due to the only small sample, the calculated risk of malignancy cannot be considered statistically significant.

Since cancer prevalence of M-TIRADS 3 was slightly higher than usual, the subgroup of seven malignant lesions of this class was reviewed. As shown in Table 4, the US presentation and FNA outcome of these cases was quite heterogeneous.

Following these ambiguous results, we retrospectively re-evaluated echogenicity of all 452 nodules initially classified as M-TIRADS 3. In 18 cases, images were not available. Among the other 434, there were 37.2% isoechoic nodules, 52.2% hypoechoic nodules, and 6.6% hyperechoic nodules. Only one (0.23%) nodule had malignant cytology and it was hypoechoic. Two isoechoic nodules and four hypoechoic nodules had cytology result suspicious for malignancy. There were no malignant or suspicious nodules in the hyperechoic nodule group. The calculated OR for isoechoic and hypoechoic nodule groups were 0.6 (95% CI 0.1164–2.9977) and 2.4 (95% CI 0.4651–11.9950), respectively; although relative risk was slightly higher in the hypoechoic nodule group, it was not statistically significant. The histology results showed a completely different prospect—in 4/7 malignant cases (57%), the nodules were isoechoic, and cytology reported benign results. In 37 histologically benign nodules out of the M-TIRADS 3 group, 35.1% were hypoechoic, 46% were isoechoic, and 2.7% were hyperechoic.

## 4. Discussion

Thyroid nodules are a common occurrence. In particular, in regions with endemic hypothyreotic goiter, the incidence of benign, asymptomatic nodules is high and becomes a diagnostic dilemma since malignancy is still present and has to be excluded. Ultrasound examination is the work horse and first choice modality in the diagnosis of thyroid nodules; therefore, appropriate criteria are needed to reduce the number of false positive findings in first-line treatment. Several standardized systems for reporting suspicious US features have been developed to enhance the precision of US in predicting thyroid nodule malignancy risk. In particular, a high negative predictive value is needed to reduce overdiagnosis and unnecessary invasive procedures. Currently, the most frequently used classifications are the 2015 American Thyroid Association guidelines [1] and different variations of TIRADS (Korean TIRADS [20], TIRADS developed by Russ et al. [21] or by Kwak et al. [13], and the recently proposed EU-TIRADS [22]). With the exception of the new EU-TIRADS, the accuracy of different systems in various studies ranged from 41.8% for TIRADS developed by Kwak to 69.5% for the Korean K-TIRADS [23,24]. The main difference between these systems is in attribution of thyroid nodule to the high-risk groups. In group 4 nodules classified by K-TIRADS and TIRADS (Russ or Kwak), the estimated malignancy risk ranged from 3.3 to 72.4%; in EU-TIRADS, it was only 6–17%. One possible explanation is that EU-TIRADS group 4 includes all mildly hypoechoic nodules without any other suspicious ultrasound features. Obviously, the criterion of hypoechogenicity on its own is not highly predictive for malignancy. Therefore, it was the aim of our study to test how far the further categorization of hypoechogenic nodules in marked and non-marked hypoechogenic lesions and classification of non-markedly hypoechogenic nodules without any other suspicious features as grade 3 would improve the overall accuracy of the modified M-TIRADS system. As hypothesized, categorization of non-markedly hypoechogenic nodules as group 3 resulted in an overall accuracy of the M-TIRADS classification of 78.26%, which was markedly higher compared to the results after application of Kwak-TIRADS 48.11%.

To classify non-marked hypoechoic nodules as TIRADS 3 improved the predictive value of group 4 for malignancy, but it did not affect the malignancy rates of the other groups. These were finally comparable to other TIRADS systems with a malignancy risk of TIRADS 3 nodules of 0.9%, TIRADS 4 nodules of 6.5–37.5%, and TIRADS 5 nodules of 82.1%.

Our results showing low risk of malignancy in TIRADS 3 group can be explained by the calculation based on cytology results. Risk of malignancy based on histology results slightly differed—the TIRADS 3 category showed 17.7% risk; therefore, we retrospectively reviewed echogenicity of all nodules in this category. The relative risk was slightly higher for non-marked hypoechoic nodules, but not statistically significant. It should be noted that most of the TIRADS 3 malignant nodules in histology are isoechoic and all of them were in Bethesda Class II, although half of the cases (two) were microfollicular carcinomas.

2D-SWE elastography sensitivity and specificity has been evaluated in a recent meta-analysis [25] for differential diagnosis of benign and malignant thyroid nodules; the results were 0.66 (95% confidence interval (CI): 0.64–0.69) and 0.78 (CI: 0.76–0.80), respectively. The application of the elastography as one of the criteria of malignancy in TIRADS during the period of this study has not yet been established. However, we recommend that examination of the thyroid gland needs to include every available modality, not only B-mode image [26], as well as taking into account other risk factors, such as age, sex and the presence of a single nodule vs. multinodular goiter, not only B-mode image [27].

The goals of the TIRADS systems are to predict malignancy risk and reduce the count of thyroid FNA biopsies, but by integration of non-marked hypoechoic nodules (what is a large part of nodules), these goals cannot be fully achieved. It has been already discussed that different US features have various specificities and sensitivities [28], and malignancy risk estimated by US is not determined by a single US feature but by a combination of coexisting US features. Nodule solidity and echogenicity are less specific features and must be found together with microcalcifications, spiculated/microlobulated margins, shape of nodule, or marked hypoechogenicity [29]. There is a higher likelihood of malignancy in TIRADS 3 category if the nodule is hypoechogenic and considered suspicious by cytology results, although isoechogenicity and benign cytology as combinations of patterns unfortunately does not exclude malignancy (three out of seven malignant nodules in M-TIRADS 3 group were isoechoic). Therefore, upgrading classification of non-marked hypoechoic nodules raises the number of unnecessary FNA biopsies and thereby the cost to the patient and healthcare systems, increasing patient and clinician anxiety. It needs to be emphasized that cost analysis was not an aim for this study, and biopsy should be done according to the indications in every case individually.

Comparing the results of our M-TIRADS and Kwak TIRADS where nodule non-marked hypoechogenicity is a suspicious feature, we found that the risk of malignancy was lower for Kwak TIRADS scoring in all TIRADS groups within this cohort, even lower than the first reported results of the Kwak study [10]. Adding hypoechogenicity as a suspicious ultrasound feature did not have a significant impact on the M-TIRADS 3 group malignancy risk (0.9% vs. 0.4%) but markedly increased malignancy risk in M-TIRADS 4 subcategories and increased the accuracy of the TIRADS scoring (M-TIRADS vs. Kwak TIRADS—78.26% vs. 48.11%, respectively).

Our study has some limitations. First, some patients were excluded from calculations because of a non-conclusive cytology. It may be discussed as a selection bias, which links with the fact that conclusive cytology may also present with more conclusive ultrasound imaging features—in particular, that a clear appearance on ultrasound improves the precision of biopsy. Second, each nodule evaluation was performed by a single radiologist, and therefore this study had a higher risk of inter-observer variability that was not a focus of this study analysis. This was reported to be a significant pitfall of US evaluation, even if improved by any TIRADS systems [30]. This limitation was addressed in this study by a consensus interdisciplinary meeting to achieve uniform methodology among performers (see Section 2.1), and therefore also no comparison of different cytopathologists’ performances was analyzed. These limitations should be taken into account for further prospective studies.

One of the causes of malignant nodule small sample size is related to Bethesda V group exclusion. Part of these nodules are malignant, but the major part is benign; therefore, the inclusion in US pattern evaluation would bring uncertainty. Using reference histology after surgery would resolve this limitation. The reason for the fine performance of M-TIRADS could be low sample size of Bethesda VI, and further larger cohort studies would be suggested with reference to post-surgery histology.

Non-marked hypoechogenicity as a malignancy sign of relatively lower value has been described in several TIRADS systems: ACR TI-RADS non-marked hypoechogenicity without other signs is classified as T3 (mild suspicious), EU-TIRADS as T4 (intermediate risk), and K-TIRADS as category 4 (intermediate suspicion) [31].

We did not perform the malignancy sign analysis according to size; nevertheless, it is published that most of the sub-centimeter nodules appear as hypoechoic, regardless their nature, and therefore FNAB is not indicated. ACR TI-RADS non-marked hypoechoic nodules recommended for biopsy should exceed a size of 2.5 cm, whereas K-TIRADS and EU-TIRADS should be above 1 cm in size [31].

## 5. Conclusions

In conclusion, our results confirm that the non-marked hypoechogenic nodule pattern correlates with a very low risk of malignancy. Exclusion of the non-marked hypoechogenicity from malignancy patterns increases the malignancy risks within TIRADS 4a and 4b categories by 20% and improves TIRADS classification diagnostic specificity and accuracy. Therefore, the classification of these nodules as TIRADS 3 appears appropriate with no significant malignancy risk alteration (0.4 to 0.9%) and increased the overall accuracy of the modified M-TIRADS score. This approach could contribute to reducing overdiagnosis, obviating the need for FNA in the patients with nodule non-marked hypoechogenicity. However, the exclusion of non-marked hypoechogenicity from malignancy groups may require regular follow-up in this specific TIRADS 3 subgroup. Further prospective research in a larger cohort with different clinical characteristics would be advised.

## Figures and Tables

**Figure 1 medicina-58-00257-f001:**
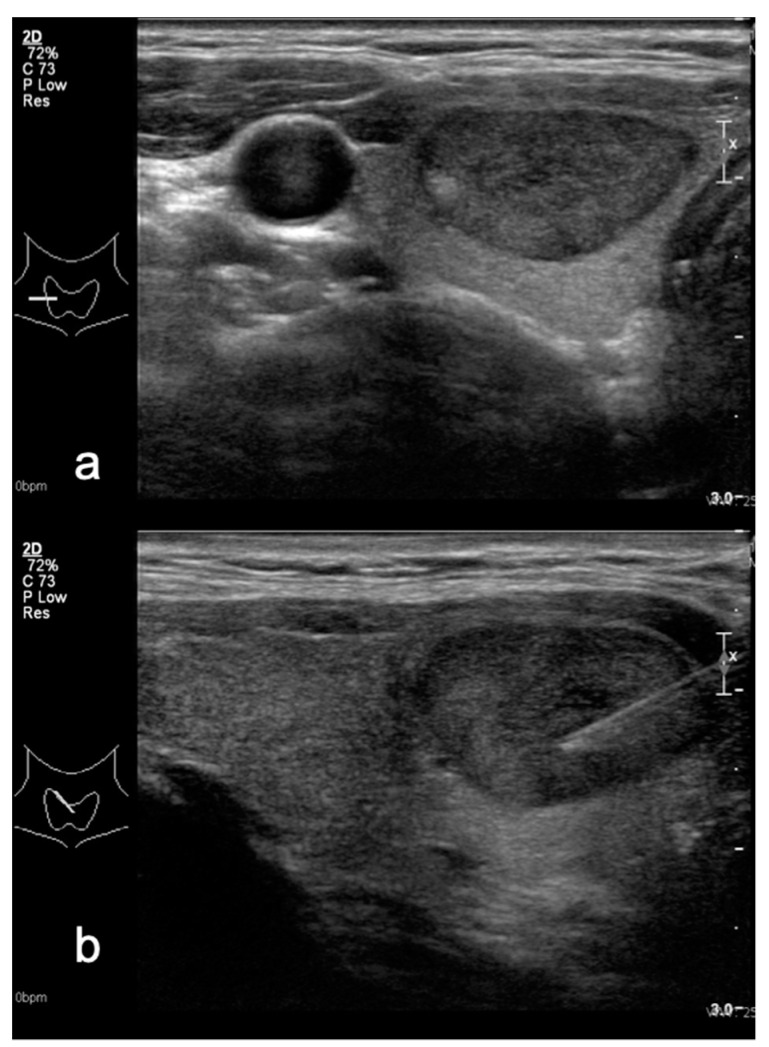
Hypoechoic thyroid nodule in (**a**) transverse and (**b**) sagittal planes—TIRADS 4A classified after Kwak system and TIRADS 3 in M-TIRADS. FNA biopsy result—Bethesda 2.

**Figure 2 medicina-58-00257-f002:**
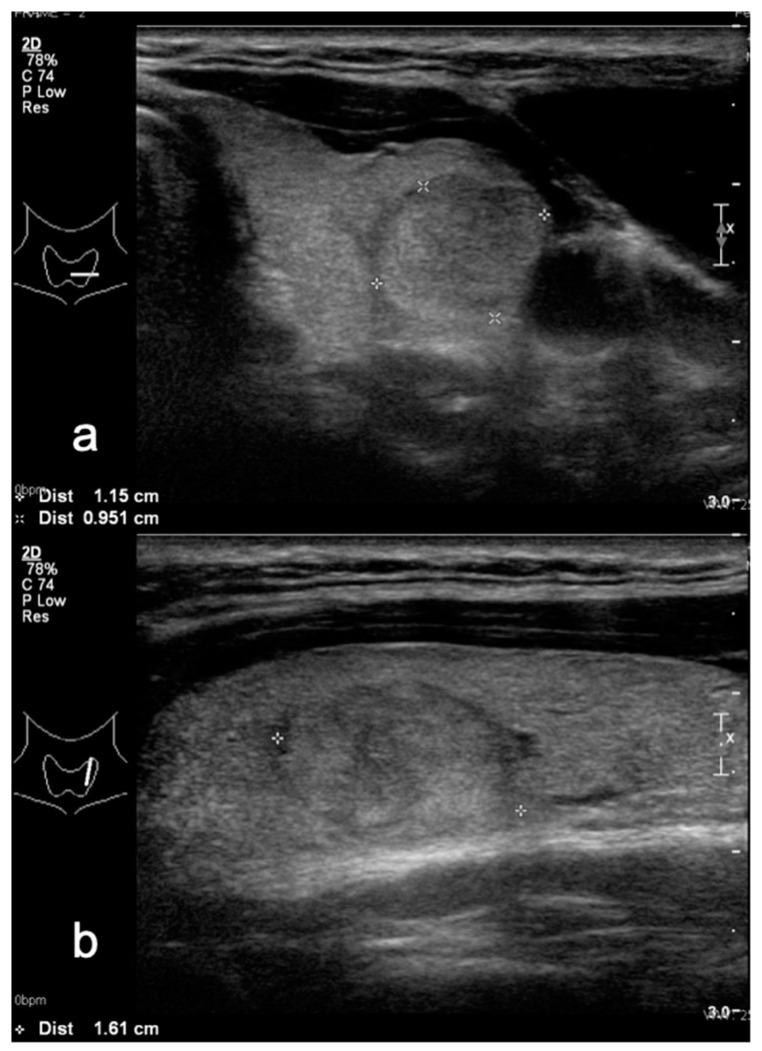
Isoechoic thyroid nodule in (**a**) transverse and (**b**) sagittal planes—TIRADS 3 classified after both Kwak and M-TIRADS. FNA biopsy result—Bethesda 2.

**Figure 3 medicina-58-00257-f003:**
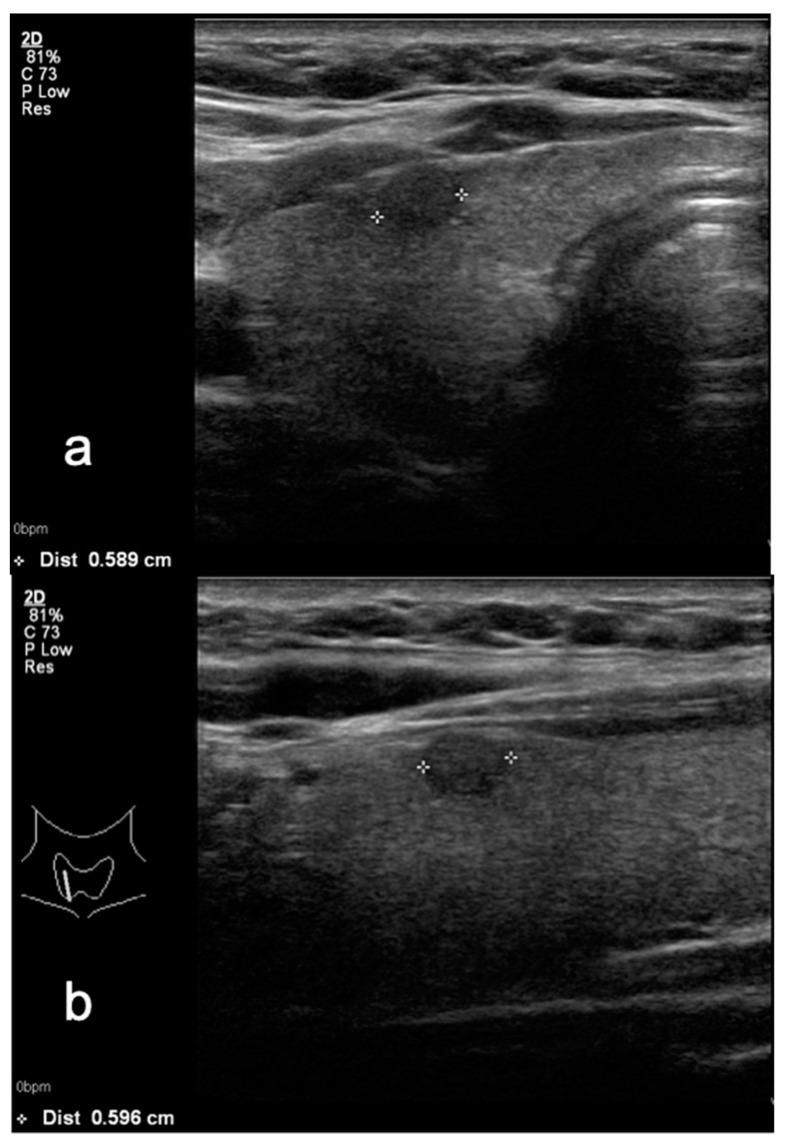
Small hypoechoic thyroid nodule in (**a**) transverse and (**b**) sagittal planes—TIRADS 4A classified after Kwak system and TIRADS 3 in M-TIRADS. FNA biopsy result—Bethesda 2.

**Figure 4 medicina-58-00257-f004:**
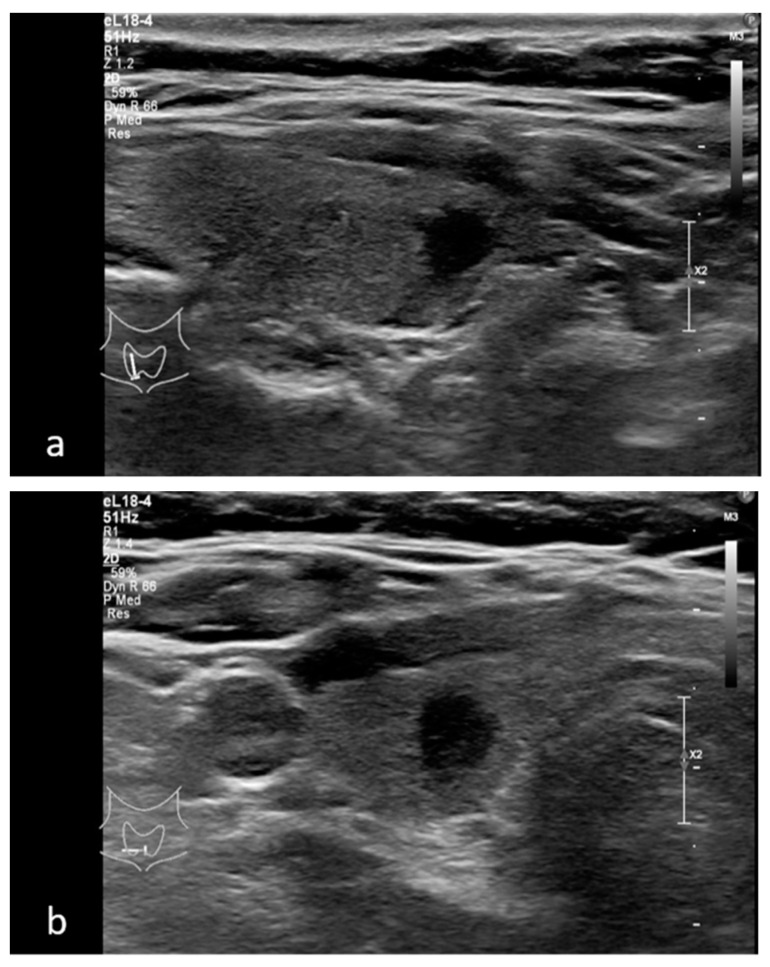
Marked hypoechoic thyroid nodule in (**a**) transverse and (**b**) sagittal planes—TIRADS 5 classified after Kwak 4C system and TIRADS 5 in M-TIRADS. FNA biopsy result—Bethesda 6.

**Figure 5 medicina-58-00257-f005:**
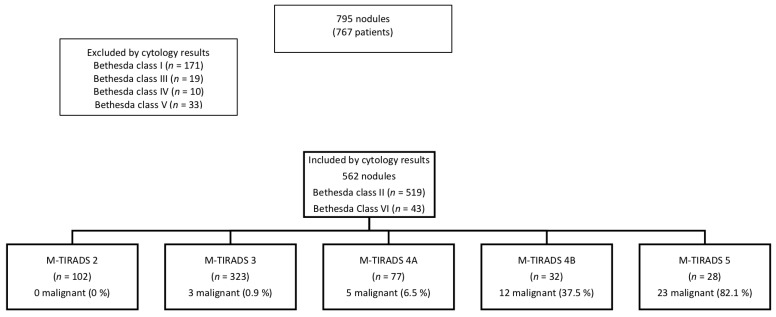
The results of M-TIRADS classification.

**Figure 6 medicina-58-00257-f006:**
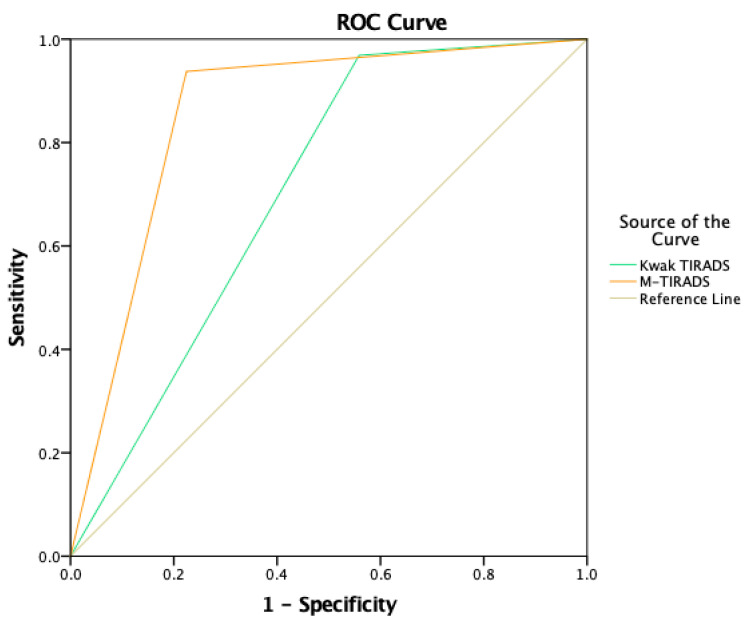
Receiver operating characteristic (ROC) curve analysis shows better specificity and sensitivity for M-TIRADS system, where non-marked hypoechogenicity is one of the signs in TIRADS 3 group, therefore downgrading the score in comparison with the Kwak TIRADS system.

**Table 1 medicina-58-00257-t001:** Summary of statistical performance of suspicious US features (*p* < 0.001).

US Feature	Sensitivity (%)	Specificity (%)	PPV (%)	NPV (%)	OR
Marked hypoechogenicity	64.86	94.32	45.28	97.37	11.43
Microcalcification	67.57	90.02	32.89	97.46	6.77
Irregular margins	75.68	93.74	46.67	98.16	12.09
AP > LL	29.73	98.63	61.11	95.05	21.07
Lymphadenopathy	19.22	100	100	94.28	177.32
M-TIRADS	93.02 (80.94–98.54)	81.31 (77.69–84.57)	29.2 (21.75–37.57)	99.29 (97.95–99.85)	18.62

AP—anterior posterior dimension, LL—latero lateral dimension.

**Table 2 medicina-58-00257-t002:** Comparison of TIRADS on the basis of Kwak and M-TIRADS.

Categories	Based on TIRADS by Kwak (*n* = 424)	Based on M-TIRADS (*n* = 460)
TIRADS categories	-	-
TIRADS 3 benign/malignant	172/1	320/3
TIRADS 4A benign/malignant	168/2	72/5
TIRADS 4B benign/malignant	42/11	20/12
TIRADS 4C benign/malignant	9/7	-
TIRADS 5 benign/malignant	0/12	5/23
Risk of malignancy	-	-
TIRADS 2	0	0
TIRADS 3	0.4	0.9
TIRADS 4A	0.8	6.5
TIRADS 4B	11.7	37.5
TIRADS 4C	23.3	-
TIRADS 5	75	82.1

**Table 3 medicina-58-00257-t003:** Comparison of primary cytology and final histology results in surgically treated patients.

M-TIRADS	Cytology		Histology	
	Bethesda Category	Cases	Category	Cases
M-TIRADS 2	Benign	7	Benign	6
			Lymphoma	1
M-TIRADS 3	Nondiagnostic	7	Benign	7
	Benign	25	Benign	22
			Thyroiditis	1
			Follicular carcinoma	2
	Follicular pathology	4	Benign	2
			Papillary carcinoma	2
	Follicular neoplasia	1	Benign	1
	Suspicious	3	Papillary carcinoma	2
			Thyroiditis	1
	Malignant	1	Papillary carcinoma	1
	No cytology	3	Benign	3
M-TIRADS 4A	Nondiagnostic	3	Benign	2
			Papillary carcinoma	1
	Benign	5	Benign	5
	Follicular pathology	1	Benign	1
	Follicular neoplasia	1	Papillary carcinoma	1
	Suspicious	5	Benign	1
			Papillary carcinoma	3
			Medullar carcinoma	1
	Malignant	4	Benign	2
			Papillary carcinoma	2
M-TIRADS 4B	Nondiagnostic	2	Benign	2
	Benign	2	Benign	2
	Follicular pathology	1	Benign	1
	Follicular neoplasia	2	Papillary carcinoma	1
			Follicular carcinoma	1
	Suspicious	6	Benign	3
			Papillary carcinoma	2
			Thyroiditis	1
	Malignant	10	Papillary carcinoma	10
	No cytology	1	Benign	1
M-TIRADS 5	Nondiagnostic	2	Benign	1
			Anaplastic carcinoma	1
	Benign	2	Benign	2
	Suspicious	5	Papillary carcinoma	4
			Thyroiditis	1
	Malignant	15	Benign	2
			Papillary carcinoma	10
			Medullar carcinoma	3
	No cytology	1	Papillary carcinoma	1

**Table 4 medicina-58-00257-t004:** Summary of M-TIRADS 3 category malignant nodules.

Nr.	Bethesda Category	US Echogenicity	Histology Result
1.	Follicular pathology	Hypo	Papillary carcinoma (follicular)
2.	Suspicious	Hypo	Papillary carcinoma
3.	Malignant	Hypo	Papillary carcinoma (follicular), multifocal
4.	Benign	Iso	Microfollicular carcinoma
5.	Benign	Iso	Papillary carcinoma (follicular)
6.	Benign	Iso	Microfollicular carcinoma
7.	Not known	Iso	Papillary carcinoma

## Data Availability

Data are available from the corresponding author after appropriate review.

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
