# Peer review of "Non-Marked Hypoechogenic Nodules: Multicenter Study on the Thyroid Malignancy Risk Stratification and Accuracy Based on TIRADS Systems Comparison"

_medicina, 2022, doi:10.3390/medicina58020257_

Round 1
Reviewer 1 Report
Although the manuscript underwent significant scientific semantic changes for the better, phraseology needs to be improved - maybe shorter sentences would make the text easily understandable.
Reviewer 2 Report
The authors satisfactorily address every comments that I have suggested. With the help of a native English speaker, the authors edited and improved the manuscript's English. Similarly, authors improved the presentation quality by providing an ROC curve, which clearly illustrates their key findings. I strongly recommend this manuscript for publication.
Reviewer 3 Report
I would like to highligth the effort made by the authors to respond to the comments submitted earlier. In my opinion, the manuscript has improved considerably and now includes important information for the interpretation of the results presented. For these reasons I believe that the article can be accepted.
Author Response
Please see the attachment.

This manuscript is a resubmission of an earlier submission. The following is a list of the peer review reports and author responses from that submission.
Round 1
Reviewer 1 Report
The subject of the article is very similar to the article of most of presented authors of the article: Impact of the Hypoechogenicity Criteria on Thyroid Nodule Malignancy Risk Stratification Performance by Different TIRADS Systems by Nina Malika Popova, Maija Radzina, Peteris Prieditis, MaraLiepa, Madara Rauda and Kaspars Stepanovs, published in Cancers 2021, 13, 5581. https://doi.org/10.3390/cancers13215581 , Published: 8 November 2021. However, the number of the patients included in the research is two times larger then th similar article.
The results are interpreted appropriately. Conclusions seem to be justified and supported by the results, although they neglect a small number of patients that could suffer from implementing the conclusions (small number of malignant nodules .that would not be confirmed if the conclusions were followed).
The article is written in an appropriate way, and the the data and analyses are presented appropriately
The study is correctly designed as possible because there is not a clear border between isoehogenic, mild hypoehogenic, hypoehogenic and marked hypoechogenic nodules.
Conclusions are interesting for the readership of the Journal and the paper could attract readership and start discussion.
This work is worth publishing, as an attempt to find more precise method of diagnosing malignant in the majority of benign thyroid noduls.
English language in the article is appropriate and understandable.

Reviewer 2 Report
This is an interesting paper that describes the study carried out to evaluate diagnostic accuracy. Considering the issues related to the difficulty of obtaining an accurate correct diagnosis, the study offers new information that may give a positive contribute to this area. The manuscript is well structured and well written which make it easy to follow.
The cited references are quite recent (approx. 2/3 were published in the last 5 years and only about 25% was published more than 10 years ago).
Some questions regarding the manuscript are presented so that the authors may detail the information.
Materials and Methods
Section 2.3: What is the influence of using different US systems or due to the difference in the professional experience of the radiologists who carried out the exams? Did you consider these parameters in the statistic analysis that was carried out? Although this is briefly referred in the discussion section as a limitation of the study, authors should consider the possibility of re-evaluating the results asking different radiologists to evaluate the samples or, at least, to carry out statistical analysis to compare the results obtained in the 3 different medical institutions where the study was carried out.
Section 2.5: Samples were analyzed by different specialized cytopathologists? As happens with the radiologists, did you evaluate how this could have affected the results? Did the authors considered the possibility of having the same sample analyzed by more than one specialist? Authors should carry out additional statistical analysis to evaluate this parameter.
Results
Lines 183-184: This sentence is confusing. Which samples (nodules) were re-classified and why? Who did this re-classification? Was the same person that did the initial classification?
Discussion
Line 273: “Elastography has been proved to have a decent sensitivity…” What do the authors mean by “decent”? Is this an objective criterion? An “decent” comparing to what?
Conclusions
What is the risk associated with applying these findings in the clinical practice based on these results? Is it necessary to carry out similar evaluations with larger populations or with patients with different characteristics/clinical history? What care should be considered before using the proposed methodology? This must be clearly stated in the conclusions.
Reviewer 3 Report
Nodules in thyroid gland is a common occurrence and can be an indication of malignancy. However, the incidence of asymptomatic and benign nodules is frequent in patients with endemic hypothyroid goiter. Hence, the detection of tumor in thyroid gland is a diagnostic dilemma. To avoid the risk of malignancy and to reduce the number of biopsies, the Thyroid Imaging Reporting and Data System (TIRADS) has been introduced in 2009 as a scoring system for systemic evaluation of thyroid malignancy. The authors evaluates the diagnostic accuracy of current TIRAD system and demonstrate that categorizing “non-marked hypoechogenic nodules” as Grade 3 nodule significantly enhance the predictive value of TIRADS classification. It is worth publishing this finding in this journal because it has clinical implications. The findings are original, have scientific soundness, and have significant content. However, the quality of presentation could be improved. I have two concerns, which should be addressed before publishing this article.
- The quality of the data presentation should be improved. For instance, the marked difference in the diagnostic accuracy of M-TIRAD in comparison to r Kwak TIRADS can be presented as a graph. Such modifications in data presentation can convey the message of the manuscript more effectively. For example, please see the sentence in the result starting as follows:
“Both TIRADS systems were analysed by Chi-square test - for M-TIRADS was 86.25 192 and for Kwak TIRADS 45.23 (p < 0.001), showing statistically significant marked difference 193 and weak correlations between cytology results for M-TIRADS (rs 0.42) and Kwak TI- 194 RADS (rs 0.22), p= 0.0001. Area under the curve was 0.70 (95% CI 0.634 – 0.776) for Kwak 195 TIRADS and 0.86 (95% CI 0.800 – 0.913) for M-TIRADS in ROC curve analysis, p=0.0001, 196 (Figure 1), which indicated that only one US sign - non-marked hypoechogenicity based 197 different grading in TIRADS system, was able to improve accuracy.”
It is possible to present these results as graphs to make them easier to understand.
- The English used in the manuscript can be improved with the help of a Native English speaker. For instance, the sentence (line 234, Section in discussion) ‘In particular in regions with endemic hypothyreotic goiter, the incidence of benign, asymptomatic nodules is high and becomes
a diagnostic dilemma, since malignancy is still present and has to be excluded’ can be rewritten as 'Particularly in regions with endemic hypothyreotic goiter, the incidence of benign asymptomatic nodules is high, which creates a diagnostic dilemma, since malignancy can still be present and needs to be ruled out.'
Such minor changes in English could enhance the readability of this manuscript.
Reviewer 4 Report
The text requires only minor english proofreading - example: using adverbs where necessary. The study design is sound and the limitations of the study are discussed. I think that the authors should emphasize in the conclusion section the observed rate of malignancy for 4B nodules according to M-TIRADS.